

# Examining abnormal Silurian trilobites from the Llandovery of Australia

Russell D.C. Bicknell[1] and Patrick M. Smith[2,3]

[1] Palaeoscience Research Centre, School of Environmental and Rural Science, University of New England, Armidale, New South Wales, Australia
[2] Department of Biological Sciences, Macquarie University, Sydney, New South Wales, Australia
[3] Palaeontology Department, Australian Museum Research Institute, Sydney, New South Wales, Australia

## ABSTRACT

Abnormal trilobites present insight into how arthropods with fully biomineralised exoskeletons recovered from injuries, genetic malfunctions, and pathologies. Records of abnormal Silurian trilobites in particular show an abundance of specimens with teratologies and a limited record of injuries. Here we expand the record of abnormal Silurian trilobites by presenting seven new abnormal specimens of *Odontopleura (Sinespinaspis) markhami* from the early Silurian (Llandovery, Telychian) Cotton Formation, New South Wales. We use these specimens to illustrate novel evidence for asymmetric distribution of pleural thoracic spine bases. These abnormal bases likely reflect genetic complications, resulting in morphologies that would unlikely have aided the fitness of abnormal individuals. In considering records of malformed Silurian trilobites more broadly, we propose that the largest trilobites may have been prey at this time. This indicates a possible change in the trophic position of trilobites when compared to Cambrian and Ordovician palaeoecosystems.

## INTRODUCTION

Abnormal extinct organisms allow for predator–prey interactions, genetic malfunctions, and injury recovery to be assessed in fossil groups (*Owen, 1985*; *Babcock, 1993a*; *Babcock, 2003*; *Babcock, 2007*; *Kelley, Kowalewski & Hansen, 2003*; *Huntley, 2007*; *Klompmaker & Boxshall, 2015*; *Leung, 2017*). Due to the palaeobiological importance of these specimens, abnormalities have been documented in many fossil groups (*Klompmaker et al., 2019*). Euarthropods, in particular, have been documented showing injuries (*Owen, 1985*; *Bicknell & Paterson, 2018*), pathologies (*Lochman, 1941*; *Šnajdr, 1978b*), and teratologies (*Pocock, 1974*; *Lee, Choi & Pratt, 2001*; *Bicknell & Smith, 2021*). While abnormalities are known from arachnids (*Mitov, Dunlop & Bartel, 2021*), crustaceans (*Bishop, 1972*; *Klompmaker et al., 2013*; *Klompmaker et al., 2014*), and horseshoe crabs (*Bicknell, Pates & Botton, 2018*), the most well documented abnormal euarthropods are trilobites (*Šnajdr, 1978a*; *Owen, 1983*; *Owen, 1985*; *Babcock, 1993a*; *Babcock, 2003*; *Fatka, Budil & Grigar, 2015*; *Fatka, Budil & Zicha, 2021*; *Bicknell, Paterson & Hopkins, 2019*; *Bicknell & Holland, 2020*; *Zong, 2021*). The detailed record of trilobite abnormalities is due to the biomineralised dorsal exoskeleton

Corresponding author
Russell D.C. Bicknell,
rdcbicknell@gmail.com

exhibited by the group, a structure that increases the preservational potential of specimens and readily permits the record of abnormal structures. Trilobites are, therefore, an ideal group for understanding how a wholly extinct clade of euarthropods experienced and recovered from abnormalities.

A large number of documented abnormal trilobite specimens are from Cambrian-aged deposits (*e.g.*, *Owen, 1985*; *Babcock, 1993a*; *Babcock, 2003*; *Pates et al., 2017*; *Pates & Bicknell, 2019*; *Bicknell & Pates, 2020*; *Zong, 2021*). These specimens commonly record failed predation (*Rudkin, 1979*; *Babcock, 1993a*; *Bicknell & Paterson, 2018*), and show limited evidence for genetic or teratological complications (see *Bergström & Levi-Setti, 1978*; *Bicknell et al., 2022a*). By contrast, the record of abnormal post-Cambrian trilobites shows developmental malformations, teratologies, and pathologies, with fewer injuries derived from predation (*e.g.*, *Owen, 1985*; *Rudkin, 1985*; *Zong, 2021*; *Bicknell et al., 2022c*). Silurian-aged deposits in particular preserve a diverse array of abnormal taxa across at least ten families (Table 1). These abnormalities primarily reflect developmental malfunctions (*Šnajdr, 1981a*; *Bicknell & Smith, 2021*), injuries and abnormal recovery from moulting (*Šnajdr, 1981a*), with rarer evidence for failed attacks (*Chinnici & Smith, 2015*; *Bicknell, Paterson & Hopkins, 2019*) and accidental trauma (*Rudkin, 1985*). These specimens also present insight into how the occasionally ornate, often iso- to macropygous, Silurian taxa recovered from moulting and developmental complications. Historically, most abnormal Silurian trilobites are reported from deposits in the Czech Republic (*e.g.*, *Přibyl & Vaněk, 1962*; *Přibyl & Vaněk, 1986*; *Šnajdr, 1976*; *Šnajdr, 1978a*; *Šnajdr, 1978b*; *Šnajdr, 1979*; *Šnajdr, 1980*; *Šnajdr, 1981a*; *Šnajdr, 1981b*), Sweden (*e.g.*, *Ramsköld, 1983*; *Ramsköld, 1984*; *Owen, 1985*; *Ramsköld et al., 1994*), and the USA (*e.g.*, *Campbell, 1967*; *Whittington & Campbell, 1967*; *Holloway, 1980*; *Rudkin, 1985*; *Whiteley, Kloc & Brett, 2002*; *Chinnici & Smith, 2015*; *Bicknell, Paterson & Hopkins, 2019*). However, more recent records of abnormal Silurian trilobites from Australia (*Bicknell & Smith, 2021*) and China (*Zong et al., 2017*; *Zong, 2021*) suggest a more Gondwanan presence of these abnormal specimens. This indicates that abnormal trilobites from middle Paleozoic may have a much more global record than previously thought. To expand this line of enquiry, here we considered the trilobite-rich Cotton Formation, central New South Wales (NSW) and illustrate new examples of abnormal odontopleurids (*Edgecombe & Sherwin, 2001*; *Rickards, Wright & Thomas, 2009*; Figs. 1 and 2).

## METHODS

Trilobite specimens from the Cotton Formation housed within the Australian Museum (AM F), Sydney, NSW, Australia were examined under a microscope. Seven abnormal *Odontopleura* (*Sinespinaspis*) *markhami* (*Edgecombe & Sherwin, 2001*) specimens were identified. These specimens were dyed black with ink, coated in magnesium oxide, and photographed under low angle LED light with a Canon EOS 5DS. An additional 39 standard specimens were also photographed using this equipment. However, as they are not figured, they were not dyed or coated. Images were stacked using Helicon Focus 7 (Helicon Soft Limited) stacking software.

Bicknell and Smith (2022), *PeerJ*, DOI 10.7717/peerj.14308

**Table 1  Record of abnormal Silurian trilobites.** Ordered by stage and then genus.

| Taxon | Family | Series | Stage | Formation, country | Abnormality location | Abnormality description | Side | Citation and figure |
|---|---|---|---|---|---|---|---|---|
| *Acernaspis elliptifrons* (*Esmark, 1833*) | Lichidae | Llandovery | Aeronian | Solvik Formation, Sweden | Pygidium | Asymmetrically developed furrows | Both | Owen (*1985*, fig. 5t) |
| *Encrinurus squarrosus* *Howells, 1982* | Encrinuridae | Llandovery | Aeronian | Newlands Formation, Scotland | Pygidium | Damaged rib | Right | Howells (*1982*, pl. 8, fig. 12) |
| *Encrinurus squarrosus* | Encrinuridae | Llandovery | Aeronian | Newlands Formation, Scotland | Pygidium | Bifurcating rib | Right | Howells (*1982*, pl. 8, fig. 13) |
| *Coronocephalus* sp. | Encrinuridae | Llandovery | Telychian | Fentou Formation, China | Pygidium | Deformed, fused pygidial ribs | Right | Zong (*2021*, fig. 4D, E) |
| *Coronocephalus* sp. | Encrinuridae | Llandovery | Telychian | Fentou Formation, China | Pygidium | Truncated pygidial ribs | Right | Zong et al. (*2017*, fig. 3q); Zong *2021*, fig. 4F, G) |
| *Coronocephalus* sp. | Encrinuridae | Llandovery | Telychian | Fentou Formation, China | Pygidium | Additional pygidial rib | Right | Zong (*2021*, fig. 4H, I) |
| *Kailia intersulcata* (*Chang, 1974*) | Encrinuridae | Llandovery | Telychian | Fentou Formation, China | Thorax | Thoracic spines 2–5 truncated, U-shaped indentation | Right | Zong (*2021*, fig. 4A–C) |
| *Odontopleura* (*Sinespinaspis*) *markhami* | Odontopleuridae | Llandovery | Telychian | Cotton Formation, NSW, Australia | Thorax | Additional thoracic spine base | Right | This article, Figs. 3A and 3B |
| *Odontopleura* (*Sinespinaspis*) *markhami* | Odontopleuridae | Llandovery | Telychian | Cotton Formation, NSW, Australia | Thorax | Additional spine base and offset spine base | Right | This article, Figs. 3C and 3D |
| *Odontopleura* (*Sinespinaspis*) *markhami* | Odontopleuridae | Llandovery | Telychian | Cotton Formation, NSW, Australia | Thorax | Additional posterior pleural band spine bases | Right | This article, Figs. 4A and 4B |
| *Odontopleura* (*Sinespinaspis*) *markhami* | Odontopleuridae | Llandovery | Telychian | Cotton Formation, NSW, Australia | Thorax | Additional thoracic spine base | Right | This article Figs. 4C and 4D |
| *Odontopleura* (*Sinespinaspis*) *markhami* | Odontopleuridae | Llandovery | Telychian | Cotton Formation, NSW, Australia | Thorax | Additional thoracic spine base | Right | This article, Figs. 4E and 4F |
| *Odontopleura* (*Sinespinaspis*) *markhami* | Odontopleuridae | Llandovery | Telychian | Cotton Formation, NSW, Australia | Thorax | Additional thoracic spine base | Right | This article, Figs. 5A and 5B |
| *Odontopleura* (*Sinespinaspis*) *markhami* | Odontopleuridae | Llandovery | Telychian | Cotton Formation, NSW, Australia | Thorax | Additional posterior pleural band spine bases | Left | This article, Figs. 5C and 5D |
| *Decoroproetus corycoeus* (*Conrad, 1842*) | Proetidae | Wenlock | Sheinwoodian-Homerian | St. Clair Formation, Arkansas, USA | Thorax, pygidium | Thoracic segment 11? fused to pygidium | Right | Holloway (*1980*, pl. 3, fig. 4) |
| *Calymene frontosa* *Lindström, 1885* | Calymenidae | Wenlock | ?Sheinwoodian | Visby Beds, Sweden | Cephalon | Abnormal development of suture | Left | Owen (*1985*, fig. 5c) |
| *Arctinurus boltoni* (*Bigsby, 1825*) | Lichidae | Wenlock | Sheinwoodian | Rochester Formation, New York, USA | Pygidium | Truncated posteriormost pygidial spine, 'W'-shaped injury | Right | Rudkin (*1985*, fig. 1A, B) |
| *Arctinurus boltoni* | Lichidae | Wenlock | Sheinwoodian | Rochester Formation, New York, USA | Thorax, pygidium | Large 'U'-shaped indentation, posterior thorax, extending onto pygidium | Right | Babcock (*1993b*, p. 36, no figure number) |
| *Arctinurus boltoni* | Lichidae | Wenlock | Sheinwoodian | Rochester Formation, New York, USA | Cephalon, thorax, pygidium | 'U'-shaped indentation, cephalon; 'V'-shaped indentation thoracic segments 3–4; 'W'-shaped indentation thoracic segments 8–10 'U'-shaped indentation pygidium | Left (cephalaon, thorax) Right (pygidium) | Whiteley, Kloc & Brett (*2002* fig. 2.9B); Chinnici & Smith (*2015*, fig. 434) |
| *Arctinurus boltoni* | Lichidae | Wenlock | Sheinwoodian | Rochester Formation, New York, USA | Thorax, pygidium | Thoracic spines 1–4 truncated, 'U'-shaped indentation, truncated pygidial spines | Right (thorax) Left (pygidium) | Chinnici & Smith (*2015*, fig. 432) |
| *Arctinurus boltoni* | Lichidae | Wenlock | Sheinwoodian | Rochester Formation, New York, USA | Cephalon, thorax | 'U'-shaped indentation, posterior cephalon, single segment injury, 4th thoracic segment | Right | Chinnici & Smith (*2015*, fig. 433) |

**Table 1** (*continued*)

| Taxon | Family | Series | Stage | Formation, country | Abnormality location | Abnormality description | Side | Citation and figure |
|---|---|---|---|---|---|---|---|---|
| *Arctinurus boltoni* | Lichidae | Wenlock | Sheinwoodian | Rochester Formation, New York, USA | Pygidium | Abnormal pygidial spine | Left | Bicknell, Paterson & Hopkins (*2019*, fig. 3A, B) |
| *Arctinurus boltoni* | Lichidae | Wenlock | Sheinwoodian | Rochester Formation, New York, USA | Pygidium | Reduced pygidial spine | Right | Bicknell, Paterson & Hopkins (*2019*, fig. 3C, D) |
| *Arctinurus boltoni* | Lichidae | Wenlock | Sheinwoodian | Rochester Formation, New York, USA | Pygidium | 'U'-shaped indentation | Right | Bicknell, Paterson & Hopkins (*2019*, fig. 3E, F) |
| *Arctinurus boltoni* | Lichidae | Wenlock | Sheinwoodian | Rochester Formation, New York, USA | Pygidium | Rounded pygidial spine | Right | Bicknell, Paterson & Hopkins (*2019*, fig. 4A, B) |
| *Arctinurus boltoni* | Lichidae | Wenlock | Sheinwoodian | Rochester Formation, New York, USA | Pygidium | 'W'-shaped indentation | Right | Bicknell, Paterson & Hopkins (*2019*, fig. 4C, D) |
| *Arctinurus boltoni* | Lichidae | Wenlock | Sheinwoodian | Rochester Formation, New York, USA | Pygidium | 'W'-shaped indentation | Right | Bicknell, Paterson & Hopkins (*2019*, fig. 4E, F) |
| *Arctinurus boltoni* | Lichidae | Wenlock | Sheinwoodian | Rochester Formation, New York, USA | Thorax | Single segment injury, thoracic segment 2 | Right | Bicknell, Paterson & Hopkins (*2019*, fig. 5A, B) |
| *Arctinurus boltoni* | Lichidae | Wenlock | Sheinwoodian | Rochester Formation, New York, USA | Thorax and pygidium | Two 'V'-shaped indentations (thoracic segments 1–2; thoracic segments 7–8); pygidium slightly truncated | Right | Bicknell, Paterson & Hopkins (*2019*, fig. 6A, B) |
| *Calymene niagarensis* (*Hall, 1843*) | Calymenidae | Wenlock | Sheinwoodian | Rochester Formation, New York, USA | Thorax | 'L'-shaped indentation, thoracic segments 1–4 | Right | Chinnici & Smith (*2015*, fig. 432) |
| *Calymene* sp. | Calymenidae | Wenlock | Sheinwoodian | Rochester Formation, New York, USA | Cephalon | Borings on genal spine | Left | Whiteley, Kloc & Brett (*2002*, fig. 2.15D–F) |
| *Coronocephalus urbis* *Strusz, 1980* | Encrinuridae | Wenlock | Sheinwoodian | Walker Volcanics, Australian Central Territory, Australia | Pygidium | Bifurcated rib | Right | Strusz (*1980*, pl. 1, fig. 17) |
| *Dalmanites limulurus* (*Green, 1832*) | Dalmanitidae | Wenlock | Sheinwoodian | Rochester Formation, New York, USA | Thorax | 'U'-shaped indentation, thoracic segments 2–5 | Right | Chinnici & Smith (*2015*, fig. 437) |
| *Dalmanites limulurus* | Dalmanitidae | Wenlock | Sheinwoodian | Rochester Formation, New York, USA | Thorax | U'-shaped indentation, thoracic segments 1–3 | Right | Chinnici & Smith (*2015*, fig. 438) |
| *Dalmanites limulurus* | Dalmanitidae | Wenlock | Sheinwoodian | Rochester Formation, New York, USA | Thorax | 'U'-shaped indentations, thoracic segments 2–4 and 8–1 | Left | Chinnici & Smith (*2015*, fig. 439); Whiteley, Kloc & Brett (*2002*, fig. 2.15A) |
| *Dalmanites limulurus* | Dalmanitidae | Wenlock | Sheinwoodian | Rochester Formation, New York, USA | Thorax, pygidium | U'-shaped indentation, thoracic segments 10–11 extending into pygidium | Left | Chinnici & Smith (*2015*, fig. 440) |
| *Dalmanites limulurus* | Dalmanitidae | Wenlock | Sheinwoodian | Rochester Formation, New York, USA | Thorax | U'-shaped indentation, thoracic segments 5–11 | Left | Chinnici & Smith (*2015*, fig. 441) |
| *Dalmanites limulurus* | Dalmanitidae | Wenlock | Sheinwoodian | Rochester Formation, New York, USA | Pygidium | Terminal, medial spine missing | Midline | Whiteley, Kloc & Brett (*2002*, fig. 2.15C) |
| *Japonoscutellum* sp. | Encrinuridae | Wenlock | Sheinwoodian | Yarralumla Formation, New South Wales, Australia | Pygidium | Bifurcating axial rib | Right | Bicknell & Smith (*2021*, fig. 3b, c) |
| *Exallaspis bufo* (*Ramsköld, 1984*) | Odontopleuridae | Wenlock | Homerian | Mulde Beds, Sweden | Cranidium | Asymmetrical cranidium | Left | Ramskold (*1984*, pl. 31, fig. 1) |

**Table 1** (*continued*)

| Taxon | Family | Series | Stage | Formation, country | Abnormality location | Abnormality description | Side | Citation and figure |
|---|---|---|---|---|---|---|---|---|
| *Exallaspis bufo* | Odontopleuridae | Wenlock | Homerian | Mulde Beds, Sweden | Pygidium | Additional terminal spine | Midline | Ramskold (*1984*, pl. 31, fig. 5) |
| *Interproetus truncus* Šnajdr, *1980* | Proetidae | Wenlock | Homerian | Liten Formation, Czech Republic | Thorax | Reduced and fused pleurae | Right | Šnajdr (*1980*, pl. XLVIII, figs 1, 2) |
| *Ktenoura retrospinosa* Lane, *1971* | Cheiruridae | Wenlock | Homerian | Much Wenlock Limestone Formation, England | Pygidium | Reduced spine | Right | Lane (*1971*, pl. 6, fig. 9a, b) |
| *Odontopleura ovata* Emmrich, *1839* | Odontopleuridae | Wenlock | Homerian | Liten Formation, Czech Republic | Thorax | 'U'-shaped indentation, thoracic segments 4–8 | Right | Šnajdr (*1979*, pl. 1) |
| *Exallaspis mutica* (Emmrich, *1844*) | Odontopleuridae | Wenlock–Ludlow | — | Grünlich-Graues Graptolithengestein, Germany | Pygidium | Single spine injury | Left | Šnajdr (*1969*, pl. IV, fig. 7) |
| *Odontopleura ovata* | Odontopleuridae | Wenlock–Ludlow | — | Grünlich-Graues Graptolithengestein, Germany | Pygidium | Asymmetric medial lobe | Left | Schrank (*1969*, pl II, fig. 4) |
| *Alcymene lindstroemi* Ramsköld et al., *1994* | Calymenidae | Ludlow | Gorstian | Hemse Marl, Sweden | Cephalon | Overdeveloped glabellar region | Midline | Ramskold (*1994*, fig. 5, 9) |
| *Bohemoharpes ungula viator* Přibyl & Vaněk, *1986* | Harpetidae | Ludlow | Gorstian | Kopanina Formation, Czech Republic | Cephalon | Asymmetrical cranidial region | Right larger than left | Přibyl & Vaněk (*1986*, pl. 2, fig.1) |
| *Bohemoharpes ungula* | Harpetidae | Ludlow | Gorstian | Kopanina Formation, Czech Republic | Cephalon | Multiple neoplasms | Left | Šnajdr (*1978a*, pl. I, figs. 1–5) |
| *Bohemoharpes ungula* | Harpetidae | Ludlow | Gorstian | Kopanina Formation, Czech Republic | Cephalon | Neoplasms on genal spine | Left | Šnajdr (*1978a*, pl. I, figs. 6, 7); Šnajdr (*1990*, p. 63) |
| *Prionopeltis archiaci* (Barrande, *1846*) | Proetidae | Ludlow | Gorstian | Kopanina Formation, Czech Republic | Pygidium | Single spine injury | Right | Šnajdr (*1981a*, pl. I, fig. 1) |
| *Prionopeltis archiaci* | Proetidae | Ludlow | Gorstian | Kopanina Formation, Czech Republic | Pygidium | 'U'-shaped indentation | Right | Šnajdr (*1981a*, pl. II, fig. 2) |
| *Prionopeltis archiaci* | Proetidae | Ludlow | Gorstian | Kopanina Formation, Czech Republic | Pygidium | Fused pygidial ribs, 'W'-shaped indentation | Right | Šnajdr (*1981a*, pl V, fig. 4) |
| *Prionopeltis archiaci* | Proetidae | Ludlow | Gorstian | Kopanina Formation, Czech Republic | Pygidium | Pinched pygidial ribs | Left | Šnajdr (*1981a*, pl V, fig. 5; pl VIII, fig. 3) |
| *Prionopeltis archiaci* | Proetidae | Ludlow | Gorstian | Kopanina Formation, Czech Republic | Pygidium | Additional terminal spine | Midline | Šnajdr (*1981a*, pl VII, fig. 6) |
| *Prionopeltis archiaci* | Proetidae | Ludlow | Gorstian | Kopanina Formation, Czech Republic | Pygidium | Thin terminal spines | Midline | Šnajdr (*1981a*, pl VIII, fig. 4) |
| *Prionopeltis archiaci* | Proetidae | Ludlow | Gorstian | Kopanina Formation, Czech Republic | Pygidium | Ribs poorly developed | Right | Šnajdr (*1981a*, pl VIII, fig. 5) |
| *Prionopeltis archiaci* | Proetidae | Ludlow | Gorstian | Kopanina Formation, Czech Republic | Pygidium | Additional spine | midline | Šnajdr (*1981a*, pl VIII, fig. 6) |
| *Prionopeltis archiaci* | Proetidae | Ludlow | Gorstian | Kopanina Formation, Czech Republic | Pygidium | Additional spine | Left | Šnajdr (*1981a*, pl VIII, fig. 7) |
| *Prionopeltis archiaci* | Proetidae | Ludlow | Gorstian | Kopanina Formation, Czech Republic | Pygidium | Additional spine | Midline | Šnajdr (*1981a*, pl VIII, fig. 8) |
| *Prionopeltis dracula* Šnajdr, *1980* | Proetidae | Ludlow | Gorstian | Kopanina Formation, Czech Republic | Pygidium | Additional spines | Both | Šnajdr (*1980*, not figured) |
| *Scharyia micropyga* (Hawle & Corda, *1847*) | Aulacopleuridae | Ludlow | Gorstian | Kopanina Formation, Czech Republic | Pygidium | 'U'-shaped indentation, spine abnormally developed | Right | Šnajdr (*1981a*, pl IV, fig. 2) |
| *Scharyia micropyga* | Aulacopleuridae | Ludlow | Gorstian | Kopanina Formation, Czech Republic | Pygidium | Additional ribs | Midline | Šnajdr (*1981b*, pl. XI, fig. 1) |
**Table 1** (*continued*)

| Taxon | Family | Series | Stage | Formation, country | Abnormality location | Abnormality description | Side | Citation and figure |
|-------|--------|--------|-------|-------------------|---------------------|------------------------|------|---------------------|
| *Scharyia micropyga* | Aulacopleuridae | Ludlow | Gorstian | Kopanina Formation, Czech Republic | Pygidium | Abnormally developed interring furrows | Midline | Šnajdr (*1981b*, pl. XI, fig. 2) |
| *Scharyia micropyga* | Aulacopleuridae | Ludlow | Gorstian | Kopanina Formation, Czech Republic | Pygidium | Abnormally developed interring furrows | Midline | Šnajdr (*1981b*, pl. XI, fig. 3) |
| *Scharyia micropyga* | Aulacopleuridae | Ludlow | Gorstian | Kopanina Formation, Czech Republic | Pygidium | Abnormal axial ring | Midline | Šnajdr (*1981b*, pl. XI, fig. 4) |
| *Scharyia micropyga* | Aulacopleuridae | Ludlow | Gorstian | Kopanina Formation, Czech Republic | Pygidium | Abnormal axial ring | Midline | Šnajdr (*1981b*, pl. XI, fig. 7) |
| *Scharyia micropyga* | Aulacopleuridae | Ludlow | Gorstian | Kopanina Formation, Czech Republic | Pygidium | Poorly developed axial rings | Midline | Šnajdr (*1981b*, pl. XI, fig. 8) |
| *Sphaerexochus latifrons* An-gelin, 1854 | Cheiruridae | Ludlow | Gorstian | Hemse Marl, Sweden | Cephalon | Pathological development on free cheek | Right | Ramsköld (*1983*, pl. 19, fig. 6) |
| *Kosovopeltis nebula* Campbell, 1967 | Scutelluidae | Ludlow | Gorstian–early Ludfordian | Henryhouse Formation, Oklahoma, USA | Thorax | Overdeveloped pleurae | Right | Campbell (*1967*, pl. 2 figs 5, 6) |
| *Batocara robustus* (Mitchell, 1924) | Encrinuridae | Ludlow | Ludfordian | Black Bog Shale, New SouthWales | Thorax | Bifurcated pleural rib | Right | Strusz (*1980*, pl. 3, fig. 7) |
| *Batocara robustus* | Encrinuridae | Ludlow | Ludfordian | Black Bog Shale, New South Wales, Australia | Pygidium | Offset axial nodes | Midline | Bicknell & Smith (*2021*, fig. 2a, b) |
| *Batocara robustus* | Encrinuridae | Ludlow | Ludfordian | Black Bog Shale, New South Wales, Australia | Pygidium | Bifurcating axial rib | Left | Bicknell & Smith (*2021*, fig. 2c, f) |
| *Batocara robustus* | Encrinuridae | Ludlow | Ludfordian | Black Bog Shale, New South Wales, Australia | Pygidium | Additional axial node | Midline | Bicknell & Smith (*2021*, fig. 2d, e) |
| *Didrepanon squarrosum* | Cheiruridae | Ludlow | Ludfordian | Kopanina Formation, Czech Republic | Crandium | Asymmetric glabellar furrows | Left | Přibyl & Vaněk (*1973*, pl. I, fig. 1) |
| *Leonaspis rattei* (Etheridge & Mitchell, 1869) | Odontopleuridae | Ludlow | Ludfordian | Black Bog Shale, New South Wales, Australia | Thorax | Asymmetrical thoracic pleural spine base | Both | Bicknell & Smith (*2021*, fig. 3a) |
| *Harpidella* (*Rhinotarion*)*setosum* Whittington & Campbell, 1967 | Aulacopleuridae | Ludlow | ?Ludfordian | Hardwood Mountain Formation, Maine, USA | Cephalon | Asymmetrical cranidium | Left larger than right | Whittington & Campbell (*1967*, pl. 5, fig. 5, 6) |
| *Prionopeltis striata* Barrande, 1846 | Proetidae | Pridoli | — | Přídolí Formation, Czech Republic | Pygidium | Single spine injury | Left | Šnajdr (*1981a*, pl. I, fig. 2) |
| *Prionopeltis striata* | Proetidae | Pridoli | — | Přídolí Formation, Czech Republic | Pygidium | 'W'-shaped indentation | Left | Šnajdr *Šnajdr (1981a)*, pl. I, fig. 3) |
| *Prionopeltis striata* | Proetidae | Pridoli | — | Přídolí Formation, Czech Republic | Pygidium | Spines removed | Left | Šnajdr (*1981a*, pl. II, fig. 3) |
| *Prionopeltis striata* | Proetidae | Pridoli | — | Přídolí Formation, Czech Republic | Pygidium | 'V'-shaped indentation | Right | Šnajdr (*1981a*, pl. II, fig. 5) |
| *Prionopeltis striata* | Proetidae | Pridoli | — | Přídolí Formation, Czech Republic | Pygidium | Fused, deformed ribs | Left | Šnajdr (*1981a*, pl. III, fig. 1) |
| *Prionopeltis striata* | Proetidae | Pridoli | — | Přídolí Formation, Czech Republic | Pygidium | 'V'-shaped indentation | Left | Šnajdr (*1981a*, pl. III, fig. 8) |
| *Prionopeltis striata* | Proetidae | Pridoli | — | Přídolí Formation, Czech Republic | Cephalon | Shallow 'U'-shaped indentation in free cheek | Right | Šnajdr (*1981a*, pl. IV, fig. 5) |
| *Prionopeltis striata* | Proetidae | Pridoli | — | Přídolí Formation, Czech Republic | Pygidium | Pathological growth | Midline | Šnajdr (*1981a*, pl. IV, fig. 6); De Baets et al. (*2021*, fig. 6.2f) |
| *Prionopeltis striata* | Proetidae | Pridoli | — | Přídolí Formation, Czech Republic | Pygidium | Additional spine, posteriormost section | Midline | Šnajdr (*1981a*, pl. VII, fig. 2) |
| *Prionopeltis striata* | Proetidae | Pridoli | — | Přídolí Formation, Czech Republic | Pygidium | 'U'-shaped indentation | Midline | Šnajdr (*1981a*, pl. VII, fig. 4) |
| *Prionopeltis striata* | Proetidae | Pridoli | — | Přídolí Formation, Czech Republic | Pygidium | 'U'-shaped indentation | Midline | Šnajdr (*1981a*, pl. VII, fig. 5) |
| *Prionopeltis striata* | Proetidae | Pridoli | — | Přídolí Formation, Czech Republic | Pygidium | 'U'-shaped indentation | Midline | Šnajdr (*1981a*, pl. VIII, fig. 1) |

Bicknell and Smith (2022), *PeerJ*, DOI 10.7717/peerj.14308

**Table 1** (*continued*)

| Taxon | Family | Series | Stage | Formation, country | Abnormality location | Abnormality description | Side | Citation and figure |
|---|---|---|---|---|---|---|---|---|
| *Prionopeltis striata* | Proetidae | Pridoli | — | Přídolí Formation, Czech Republic | Pygidium | 'W'-shaped indentation | Left | Šnajdr (*1981a*, pl. VIII, fig. 2) |
| *Scharyia nympha Chlupáč, 1971* | Aulacopleuridae | Pridoli | — | Přídolí Formation, Czech Republic | Pygidium | Additional ribs, asymmetrically developed | Midline | Šnajdr (*1981b*, pl. XII, fig. 7) |
| *Tetinia minuta* (*Přibyl & Vaněk, 1962*) | Proetidae | Pridoli | — | Přídolí Formation, Czech Republic | Pygidium | Reduced ribs | Right | Šnajdr (*1981a*, pl. II, fig. 7) |
| *Tetinia minuta* | Proetidae | Pridoli | — | Přídolí Formation, Czech Republic | Pygidium | 'U'-shaped indentation, pinched ribs | Right | Šnajdr (*1981a*, pl. II, fig. 8) |
| *Tetinia minuta* | Proetidae | Pridoli | — | Přídolí Formation, Czech Republic | Pygidium | U'-shaped indentation, abnormal ribs | Left | Šnajdr (*1981a*, pl. III, fig. 4) |
| *Tetinia minuta* | Proetidae | Pridoli | — | Přídolí Formation, Czech Republic | Pygidium | Asymmetrical pygidium, abnormal ribs | Left | Šnajdr (*1981a*, pl. III, fig. 5) |
| *Tetinia minuta* | Proetidae | Pridoli | — | Přídolí Formation, Czech Republic | Pygidium | Asymmetrical medial lobe, abnormal ribs | Left | Šnajdr (*1981a*, pl. III, fig. 6) |
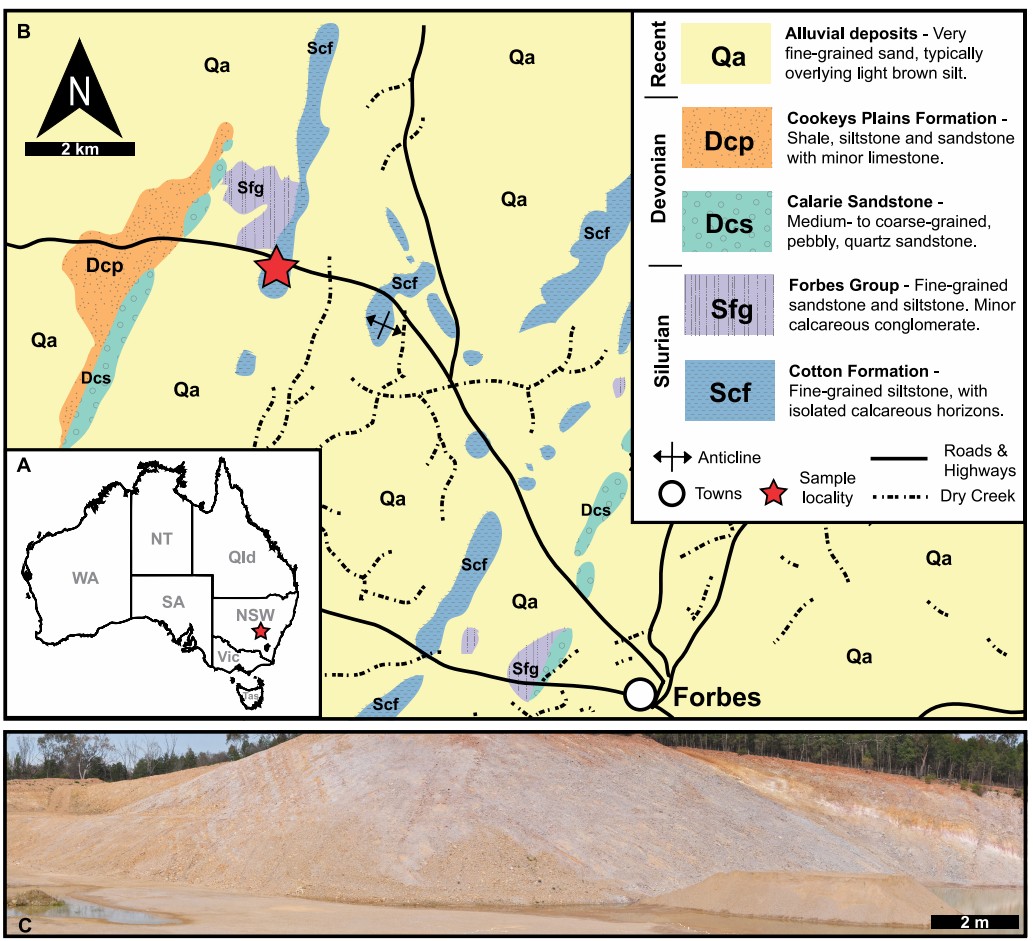

**Figure 1** **Geological, stratigraphic, and geographical information for specimen locations and the Cotton Hill Formation.** (A) Map of Australia showing specimen location (red star) in New South Wales. (B) Geological map showing rocks proximal to Forbes. Red stars indicate specimen location. (C) Panoramic view of located where specimens were collected–Cotton Hill Quarry.

A dataset of linear measurements was collated to determine where abnormal *Odontopleura* (*Sinespinaspis*) *markhami* specimens are located relative to standard individuals in bivariate space. Measurements of the cranidial length, glabellar width, and combined thorax and pygidium length were taken from 46 specimens ($n = 39$ normal, $n = 7$ abnormal) in the AM F collection (Fig. 3). The dataset was collated from specimen photographs using ImageJ (*Schneider, Rasband & Eliceiri, 2012*) (Data S1). Measurements were natural-log normalised and plotted, points were colour coded for presence or absence of abnormalities.

## Geological context

The material reported herein comes from "Cotton Hill Quarry", at approximately 33°18′44.0″S 147°56′00.9″E, on the western limb of the Forbes Anticline within the Cotton Formation (Fig. 1). The geological context of this site was discussed in detail by

| | Series | Geological Unit |
|---|---|---|
| **Silurian** | Wenlock | Forbes Group |
| | Llandovery | Cotton Hill Formation (upper) (middle) ⭐ |
| | | |
| **Ordovician** | Upper | Cotton Hill Formation (lower) |
| | | Northparkes Volcanic Group |

**Figure 2** **Correlation of selected Late Ordovician and Silurian rock units surrounding the Cotton Formation within the Forbes area.** Approximate position of sampled trilobite horizon indicated by red line and star symbol. Grey section indicates time break between the lower and two upper members (*Percival & Glen, 2007*).

Edgecombe & Sherwin (*2001*, p 87–90). Hence, only a summary is provided here. Generally, the formation outcrops poorly, appearing only as low rubbly hills in the Forbes region. Occasionally it is exposed in road and rail cuttings, as well as locally in gravel quarries. The Cotton Formation at "Cotton Hill Quarry" consists of well-bedded, thinly to moderately laminated siltstone which readily splits along the bedding plane (Fig. 1C). The outcrop varies considerably in colour, mostly being an off-white to light brownish yellow. However, in limited patches, it is deep orange to purple, often associated with large Liesegang rings. The floor of the quarry reveals that the original, unweathered rock is a darker grey colour and contains interbeds of whiter tuff that show signs of small-scale slumping. The quarry walls indicate a dip at 65° to the west and a minimum thickness of 105 m in its upper member. Previous reports suggest the entire Cotton Formation could be up to 1,500 m in total thickness on the eastern limb of the Forbes Anticline (*Sherwin, 1973*), assuming a consistent dip and no cover.

Traditionally, the entire Cotton Formation was thought to range across the Ordovician—Silurian boundary (*Sherwin, 1970*; *Sherwin, 1973*; Fig. 2). However, to date, only three horizons are known to contain age diagnostic graptolite faunas. The oldest of these—the "lower member"—has been assigned a possible Katian (late Ordovician) age. The "middle" and "upper members" contain fauna indicative of early and late Llandovery (early Silurian) age respectively (*Sherwin, 1974*; *Rickards, Wright & Thomas, 2009*). So far, there is no conclusive evidence of Hirnantian or earliest Llandovery graptolites, suggesting a significant time break between the "lower member" and the remaining two members in the formation (*Percival & Glen, 2007*). The material from "Cotton Hill Quarry" is derived from singular horizons within the upper-most 50 m of the formation, typically the "upper member". Here the trilobites co-occur with a distinct *Spirograptus turriculatus* Zone graptolite fauna. Sherwin (*1973*, fig. 4) also noted a similar trilobite fauna ~20 m from the quarry, occurring one meter above beds with the eponym of the graptolite zone. Sherwin also noted the trilobites occurred 100 m stratigraphically above a horizon with

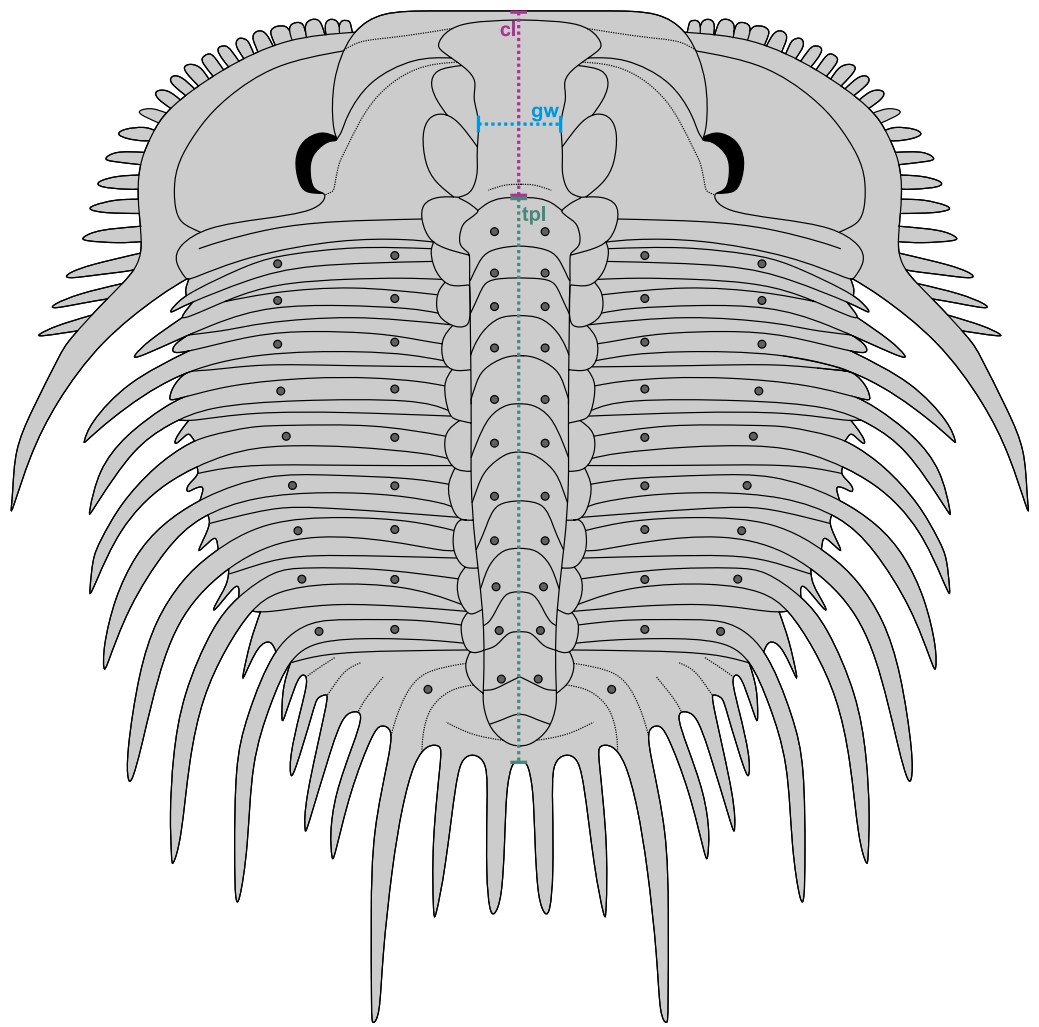

**Figure 3  Reconstruction of *Odontopleura* (*Sinespinaspis*) *markhami* showing measurements taken for analysed dataset.** Abbreviations: cl, cranidial length; gw, glabellar width; tpl, combined thorax and pygidium length.

*Monograptus* cf. *sedgwicki*. This strongly supports a late Llandovery age for the "Cotton Hill Quarry" material (*Edgecombe & Sherwin, 2001*).

Variability in lithology of the members has resulted in a variety of depositional environments suggested for the Cotton Formation (*e.g.*, *Krynen, Sherwin & Clarke, 1990*). The "upper member" exposed at "Cotton Hill Quarry" likely formed in a calm outer-shelf environment, below storm wave base, as evidenced by the well-laminated siltstone and the lack of disarticulated trilobites and echinoderms. The abundant planktonic graptolites and common small-eyed (or blind) trilobite taxa suggest that the environment was relatively deep, limiting light penetration. However, the benthic faunas (*e.g.*, rare dendroidal graptolites, strophomenid brachiopods, platyceratid gastropods, and echinoderms) suggests that the bottom waters were still well-oxygenated.

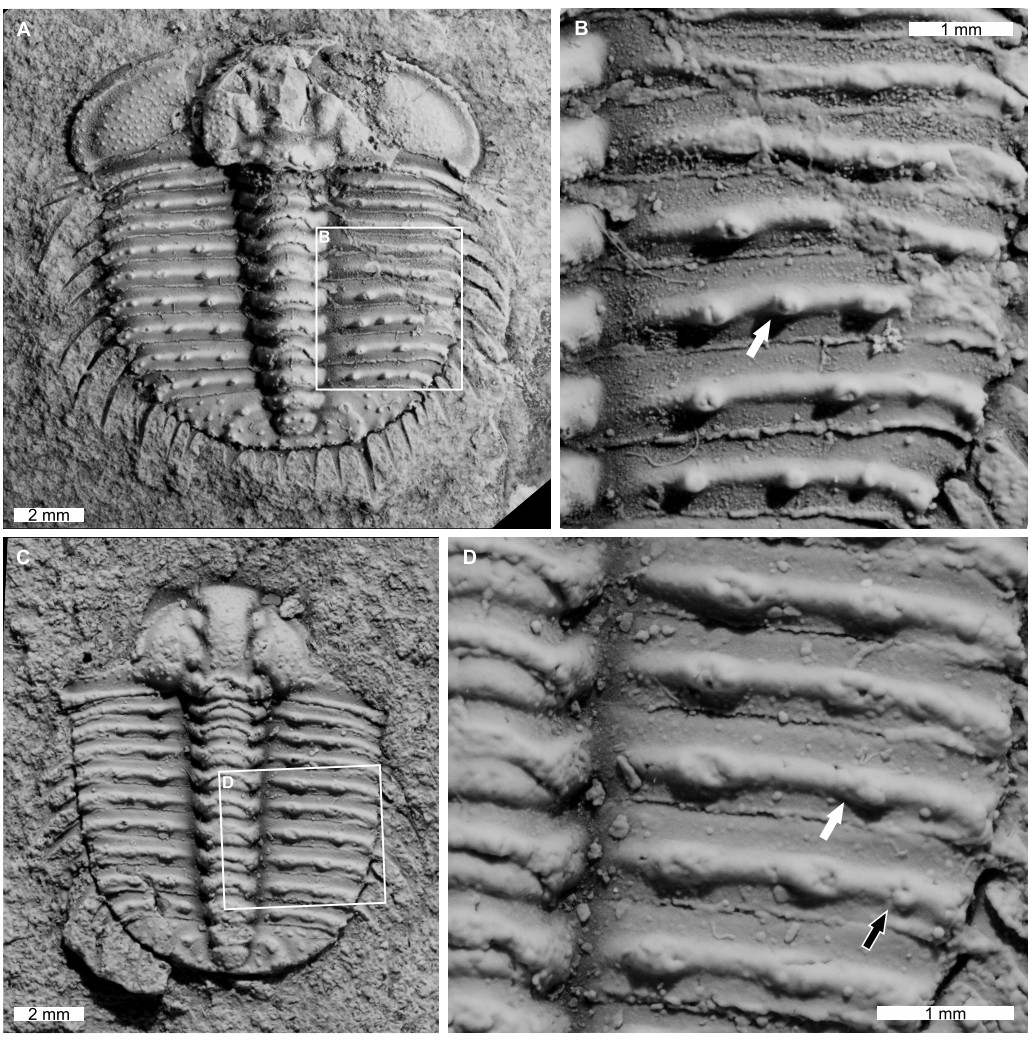

**Figure 4** *Odontopleura* (*Sinespinaspis*) *markhami* **with additional and abnormal spine bases on the right thoracic lobe.** (A, B) AM F126904. (A) Complete specimen. (B) Close up of box in (A) showing additional spine base on the seventh thoracic segment (white arrow). (C, D) AM F118762. (C) Complete specimen. (D) Close up of box in (C) showing offset spine base (white arrow) and additional spine base (black arrow).

## RESULTS

Abnormalities on *Odontopleura* (*Sinespinaspis*) *markhami* are minute (sub-millimetre scale) and primarily record the asymmetry of thoracic posterior pleural band spine bases.

AM F126904 is a near complete specimen, 13.3 mm long, 10.3 mm wide (excluding genal and pleural spines) with an asymmetric distribution of thoracic posterior pleural band spine bases (Figs. 4A, 4B). The seventh thoracic segment on the right pleural lobe has an additional spine base when compared to the left side.

AM F118762 is a moult, lacks free cheeks, is 12.2 mm long, 10.2 mm wide (excluding pleural spines) with one offset spine base and one additional spine base on the right pleural

lobe (Figs. 4C, 4D). The sixth thoracic segment has an offset spine base and the seventh segment has an additional base.

AM F115089 is a partial specimen, lacks a posterior section, is 13.3 mm long, 12.0 mm wide (excluding pleural and genal spines) with an asymmetrical distribution of thoracic posterior pleural band spine bases (Figs. 5A, 5B). The first, third, and fourth thoracic segments on the right pleural lobe have an additional spine base not observed on the left lobe.

AM F115081 is a partial specimen, lacking the posterior portion of the exoskeleton, likely a moult, is 10.8 mm long, 7.0 mm wide (excluding pleural spines). The specimen has an additional thoracic spine base on the left pleural lobe (Figs. 5C, 5D). The third thoracic segment has an additional base not observed on the right lobe.

AM F145135 is 11.7 mm long, 12.4 mm wide (excluding pleural and genal spines) with an additional thoracic spine base on the right pleural lobe (Figs. 5E, 5F). The second thoracic segment has an additional base not observed on the left lobe.

AM F118772 is likely a moult, lacks free cheeks, is 14.7 mm long, 12.9 mm wide (excluding pleural spines). The specimen has an abnormal spine base on the right pleural lobe (Figs. 6A, 6B). The sixth thoracic segment has a thoracic spine base unaligned with the immediately anterior and posterior spine bases.

AM F133034 is likely a moult, lacks free cheeks, is 10.7 mm long, 9.1 mm wide (excluding pleural spines). The specimen has an asymmetrical distribution of thoracic pleural spine bases (Figs. 6C, 6D). The sixth and eighth thoracic segments on the left pleural lobe have an additional spine bases not observed on the right lobe.

Considering the size distribution of *Odontopleura* (*Sinespinaspis*) *markhami* in bivariate space, four distinct clusters are noted (Fig. 7). We propose that four holaspid size groups are documented. The abnormal specimens are located within the second largest observed size grouping.

## DISCUSSION

*Odontopleura* (*Sinespinaspis*) *markhami* abnormalities represent additional thoracic spine base developments or offset of spine bases. Despite the presence of these abnormal structures, there is no evidence for exoskeletal removal, or any other damage to specimens. Therefore, abnormal spine base development does not reflect abnormal recovery from an injury induced during moulting or from a failed attack. These abnormalities must have arisen through another process. In life, odontopleurid trilobites had large spines that preserve as spine bases on internal moulds (*Bruton, 1966*). Additional spine bases therefore record development of spines that arose outside the primary spine sequences. Such additional spines may have resulted in more effective defence against possible predators. However, the Cotton Formation biota show few predators (*Edgecombe & Sherwin, 2001*). Furthermore, the spines would not have resulted in an increased reproductive fitness as thoracic spinosity is unlikely to be a sexually selected morphology, unlike cephalic spines (*Knell & Fortey, 2005*; *Knell et al., 2013*). Given these conditions, it seems that the additional bases record teratological developments through genetic malfunctions.

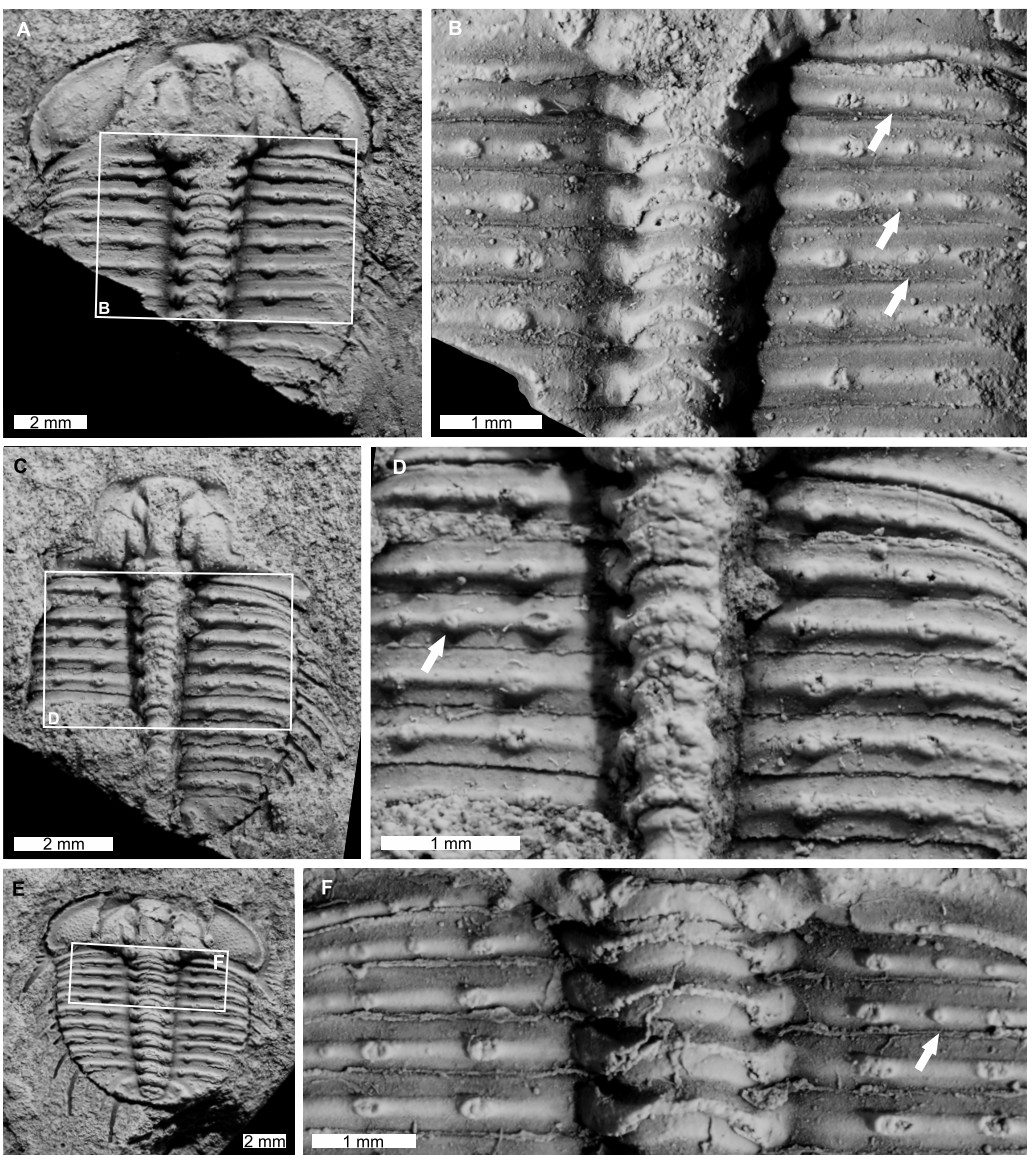

**Figure 5** *Odontopleura* (*Sinespinaspis*) *markhami* showing additional spine bases. (A) Complete specimen. (B) Close up of box in (A) showing additional spine bases on first, third, and fourth thoracic segments on the right pleural lobe (white arrows). (C, D) AM F115081. (C) Complete specimen. (D) Close up of box in (C) showing additional spine base on the third thoracic segment of the left pleural lobe (white arrow). (E, F) AM F145135. (E) Complete specimen. (F) Close up of box in (E) showing additional spine bases on second thoracic segment of the right pleural lobe (white arrow).

Similar additional spine bases were observed on a specimen of *Leonaspis rattei*—an odontopleurid from the Ludfordian Black Bog Shale, NSW (*Bicknell & Smith, 2021*, fig. 3a). These abnormal spine bases were attributed to fluctuating asymmetry—"random and uncorrelated deviations in the expression of normally bilateral characters" (*Smith, 1998*, pg. 99) indicating irregularities during the developmental processes. Although a more

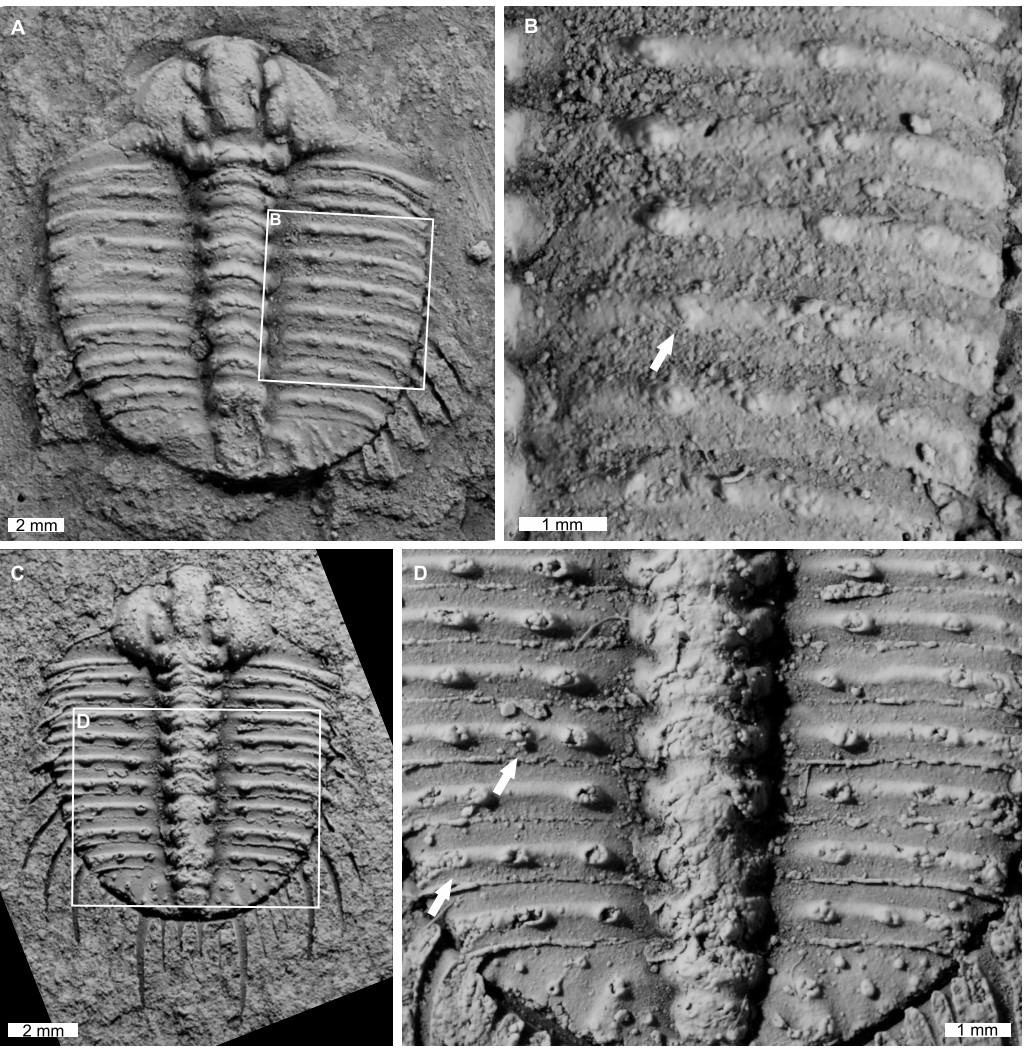

**Figure 6** *Odontopleura* (*Sinespinaspis*) *markhami* with additional and offset spine bases. (A, B) AM F118772. (A) Complete specimen. (B) Close up of box in (A) showing offset spine on the sixth thoracic segment of the right pleural lobe (white arrow). (C, D) AM F133034. (C) Complete specimen. (D) Close up of box in (C) showing additional spine bases on the sixth and eighth thoracic segments of the left pleural lobe (white arrows).

thorough examination of the Odontopleuridae is needed, these abnormal structures may be more common than previously considered.

Abnormal spines been observed in modern decapod crustaceans (*Rasheed, Mustaquim & Khanam, 2014*; *İlkyaz & Tosunoğlu, 2019*; *Waiho, Ikhwanuddin & Fazhan, 2022*) and horseshoe crabs (*Bicknell & Pates, 2019*; *Bicknell et al., 2022b*). The majority of these spines are associated with a larger injury and have therefore been attributed to complicated moulting or failed predation. However, in the rare situations where there is no evidence for injuries, possible genetic malfunctions have been presented to explain these spines (*İlkyaz*

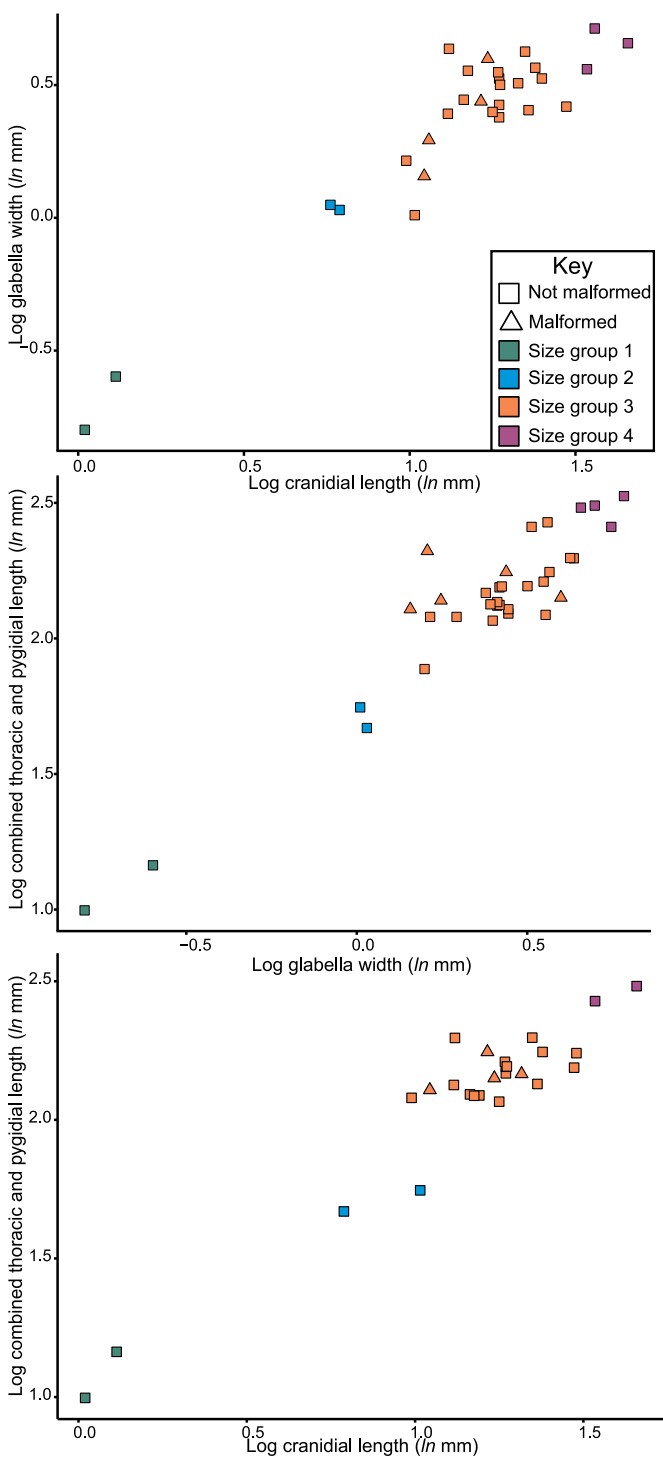

**Figure 7** **Natural log normalised bivariate plots of *Odontopleura* (*Sinespinaspis*) *markhami* of abnormal and standard specimens.** Abnormal specimens are located in size group 3.

& *Tosunoğlu, 2019*). It seems possible that trilobites with a large number of spines may have experienced malfunctions in a similar fashion to modern, spine-bearing arthropods.

The distribution of *Odontopleura* (*Sinespinaspis*) *markhami* specimens in bivariate space illustrates that most abnormal specimens are located within the second largest size grouping. This could be interpreted as evidence for an increased frequency of abnormal spines within *O.* (*S.*) *markhami* during later growth stages. However, this pattern of increased specimens is influenced by the limited sampling from other size groups and the lack of a complete ontogenetic sequence of the species. As such, the presence of abnormal specimens in all developmental stages cannot be discounted. To shed more light on the presence of abnormal spines within *O.* (*S.*) *markhami*, more specimens, and ideally a complete development sequence, are needed. Further, examining abnormality patterns within other odontopleurid species, and trilobites more broadly, using a population-based approach will uncover generalized patterns across the clade's extensive evolutionary history. However, such a collation of data was far beyond the scope of the present paper and represents important future directions for understanding abnormal specimens within trilobite populations.

Considering the record of abnormal Silurian trilobites from all parts of the globe (Table 1) most abnormal specimens record developmental complications and teratological recovery from substandard moulting (*Bicknell & Smith, 2021*), with rare examples of pathologies (*De Baets et al., 2021*). However, for the larger (>4 cm length) Silurian trilobites, such as *Arctinurus boltoni*, *Calymene niagarensis*, and *Dalmanites limulurus* from the Wenlock (Sheinwoodian) Rochester Formation, abnormalities include the removal of large exoskeletal sections (*Babcock, 1993b*; *Whiteley, Kloc & Brett, 2002*; *Chinnici & Smith, 2015*; *Bicknell, Paterson & Hopkins, 2019*). These record failed predation, as opposed to moulting complications (*Chinnici & Smith, 2015*; *Bicknell, Paterson & Hopkins, 2019*), especially as these taxa lack elongated pleural spines that would have complicated moulting (*Conway Morris & Jenkins, 1985*; *Bicknell & Pates, 2020*). The size of the species may therefore play a fundamental role in whether trilobite groups are targeted for predation. Indeed, Cambrian trilobites represented some of the largest prey items in the period and likely were targeted as food items (*Bergström & Levi-Setti, 1978*; *Holmes, Paterson & García-Bellido, 2020*; *Bicknell et al., 2022a*). The same is applicable for large, injured Ordovician species (*Bicknell et al., 2022c*; *Bicknell et al., 2022d*). As such, by the Silurian, other prey items (such as eurypterids) may have been preferred and only in select paleoecosystems were larger trilobite taxa subject to higher predation pressure. Alternatively, smaller trilobite species were completely consumed during predation, removing evidence from the fossil record. One possible means of testing this is to examine shelly coprolites from Silurian-aged deposits for trilobite fragments. Such an assessment may shed light on whether the bias for larger injured trilobites is a genuine biological signal, or the result of survivorship bias.

## ACKNOWLEDGEMENTS

We thank Matthew McCurry of the Australian Museum, Sydney for generously allowing access to the specimens under his care, and kindly allowing PMS to use the museum's

palaeobiology laboratory. We also thank Stephen Pates for discussions on earlier versions of the paper. Finally, we thank John Foster, Lisa Amati, and an anonymous reviewer, as well as the editor Bruce Lieberman, for their comments and suggested changes that improved the scope and direction of the manuscript.

### Funding

This research was funded by a University of New England Postdoctoral Fellowship (to Russell D.C. Bicknell), a Karl Hirsch Memorial Grant (to Russell D.C. Bicknell), and an Australian Museum AMF/AMRI Visiting Research Fellowship (to Russell D.C. Bicknell). The funders had no role in study design, data collection and analysis, decision to publish, or preparation of the manuscript.

### Grant Disclosures

The following grant information was disclosed by the authors:
University of New England Postdoctoral Fellowship.
Karl Hirsch Memorial Grant.
Australian Museum AMF/AMRI Visiting Research Fellowship.

### Competing Interests

The authors declare there are no competing interests.

### Author Contributions

- Russell D.C. Bicknell conceived and designed the experiments, performed the experiments, analyzed the data, prepared figures and/or tables, authored or reviewed drafts of the article, and approved the final draft.
- Patrick M. Smith conceived and designed the experiments, analyzed the data, prepared figures and/or tables, authored or reviewed drafts of the article, and approved the final draft.

### Data Availability

The raw data is available in the Supplementary File.

### Supplemental Information

Supplemental information for this article can be found online at http://dx.doi.org/10.7717/peerj.14308#supplemental-information.

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
