# Peer review of "Examining abnormal Silurian trilobites from the Llandovery of Australia"

_PeerJ, doi:10.7717/peerj.14308_

## Round 0.1 · original submission · Minor Revisions

There are a number of useful comments here and I would pay close attention to and try to address many of them. In particular, please make all of the changes suggested by reviewer's #1 (Foster) and #2 (Amati) both in their review comments and also on the changes each has provided on their annotated manuscripts. Then, regarding reviewer #3, I do think there are some useful and helpful comments and suggestions there. In particular, to try to simplify things for you and the the next stage of the process I would suggest you make these ones requested:

"adding a lithostratigraphic chart of the locality of the Cotton Formation with the horizon the specimens came from clearly marked."

and

"I would suggest that the authors delineate the clusters they see in Figure 6."

Then, there are a couple that I think you should make additions/modifications to the text of the paper to try to address. In particular, regarding this one:

"It seems to me that most of the record of abnormal Silurian trilobites contain other families that don’t have the same morphology as Odontopleurids – any chance that you could compare other odontopleurid genera/species with thoracic spines to see how prevalent these abnormalities are? A broader sampling would provide information of how common these occurrences are in the family and bolster the statistics."

I don't think you need to add more samples, but I do think adding a discussion that your sample is not complete and is based on a particular time period and set of taxa would be very worthwhile.

Then, regarding this one:

"I think that there the small dataset is doing some pretty heavy lifting in this paper. Statements like 'larger trilobite taxa were subject to higher predation pressure' and 'This illustrates are marked change in the trophic level for trilobites….' are not borne out by this small study from one locality with 46 specimens from Australia. "

I think the reviewer is correct and it would be worthwhile to add some kind of caveats to the result, something on the order of: "While these results do have potential bearing on the nature of predation on all trilobites, only a limited number of specimens from a particular region and time interval could be considered. It will be worthwhile to see to what extent these results can be generalized across the clade's extensive evolutionary history, but that is beyond the scope of the present paper." Or something to that effect. Basically this acknowledges that data are always incomplete and takes a more conservative approach, which is better justified anyways.

When you resubmit, please provide a letter detailing the changes you have made in response to the reviews, as well as the changes that you did not make. Also please provide a copy of the manuscript with the changes tracked, in addition to a clean copy.

·

Basic reporting

No comment

Experimental design

No comment.

Validity of the findings

Meets all standards, but I have two comments regarding possible additions to the Discussion:

1) According to my rough estimates from Figure 6, somewhere around 14-18% of individuals in the sample have some type of abnormality to the spine rows on at least one thoracic segment. This seems fairly high, and perhaps it is due to a relatively low sample size so far. But I wonder if there is a possibility of a higher rate of developmental or genetic abnormalities to these more "exterior" structures. Are there data on rates of such abnormalities to more "core" morphological structures in trilobites or other arthropods (e.g., eye, glabella, axial lobe, etc.), that would suggest the spines and other similar structures were more susceptible to these types of abnormalities?

2) It seems odd that larger taxa would be more frequent targets of predation. Are there specific numbers of injury rates of defined large vs. small taxa? Is it possible that smaller taxa could have simply been fully consumed more often and larger taxa are more likely to survive individually, to then demonstrate injuries as fossils? More background and data for this paragraph would be good.

Additional comments

Also have a few edits marked up in the PDF.

·

Basic reporting

The paper is well written, references are appropriate, and the context is supplied. My only suggestion is that lines 125 - 148 could be made into another table. The paper is fine either way.

Experimental design

Everything is fine.

Validity of the findings

No comment.

Additional comments

These numbered comments refer to the numbers written in the margins in pencil on the scanned copy of the paper attached.

1. In the last sentence of the abstract, you say that you are comparing Silurian trilobites to trilobites in early and middle Paleozoic ecosystems. But the Silurian is early to middle Paleozoic.

2. It sounds like you’re saying that more recent records show that there are more abnormalities in Gondawana. But maybe what you’re trying to say is that more recent, Gondwanan discoveries show that these occurrences are global.

3. It sounds like you only blackened, whitened and photographed the 7 abnormal specimens but in the next paragraph you say that you took measurements for 46 specimens from photos.

4. I don’t know what “proximal” to the quarry means.

5. Do you mean the graptolite zone?

6. Could you make this into a table with headings specimen number, length, width, molt or carcass, abnormality?

7. In the text, you interpret the cause of the abnormalities for the specimens in the table. Maybe add a column to the table with interpretation of cause of abnormality.

Reviewer 3 ·

Basic reporting

This paper is clear and well written. The literature is well referenced and there is ample background discussing abnormalities in trilobites. I am very interested in knowing more about the growth and development of these particular trilobites as it relates to the addition of thoracic segments and when these malformations may actually appear. Do these animals add a thoracic segment for each molt and therefore you can “track” the first occurrence of these malformations in development – or are the malformations added after all thoracic segments are in place and “random” spines are just added to existing thoracic segments during molting in the holaspid stage?

The figures are relevant, but I would suggest adding a lithostratigraphic chart of the locality of the Cotton Formation with the horizon the specimens came from clearly marked.

I would suggest that the authors delineate the clusters they see in Figure 6. One can see some clustering but 2 specimens in one cluster doesn't really make for a statistical argument, although the authors do acknowledge that this is the case.

Experimental design

Most of my concerns center around the number of specimens in the sample and the ability to make larger statements based upon these findings.

I understand that the authors want to link developmental stage/size to the presence of these malformations, perhaps suggesting that something in later development is responsible for these oddities in spine number and position on the thoracic segments. However, the sample size is very limited, with only 4-5 measured specimens with malformations in the graph. There are as many unmalformed specimens in the second largest developmental stage as there are malformed. I think with only 4-5 specimens representing the malformed specimens, the link to size and developmental stage is tenuous.

Validity of the findings

It seems to me that most of the record of abnormal Silurian trilobites contain other families that don’t have the same morphology as Odontopleurids – any chance that you could compare other odontopleurid genera/species with thoracic spines to see how prevalent these abnormalities are? A broader sampling would provide information of how common these occurrences are in the family and bolster the statistics.

Why would additional or offset spines be unbeneficial for individuals? Not all genetic malfunctions need to be maladaptive – it could just be neutral to the success of the individual. Neither here nor there – just an oddity. I think a stronger argument for it being unbeneficial is needed, for example - do offset and additional spines increase the risk for failure during molting?

I think that there the small dataset is doing some pretty heavy lifting in this paper. Statements like “larger trilobite taxa were subject to higher predation pressure” and “ “ This illustrates are marked change in the trophic level for trilobites….”are not borne out by this small study from one locality with 46 specimens from Australia. These conclusions are whole ecosystem findings that, while interesting, need more broad contextual data to support. I would suggest focusing on what the abnormalities in these trilobites tell us about growth and development of these animals and how this fits into the broader context of abnormalities within trilobites.

Additional comments

I think this is a very interesting study but am afraid that the conclusions may be tenuous given the small sample size of the dataset. If additions of other odontopleurid species were added, specifically looking at thoracic segment spine placement, it might provide a more robust argument for the findings. Of course, these spine abnormalities may be very rare in the family as a whole (which is an interesting point), but it may also be that this has been overlooked in other species. At the very least, it could provide more context for the significance of these specimens from Australia and provide a stronger basis for some of the arguments regarding prey size and trophic level.

---

## Round 0.2 · Minor Revisions

The authors have done a good job addressing the reviewer comments and my suggested changes and the paper is in my opinion just about ready to be accepted. The only thing that is still needed is derived from a request from the Section Editors to provide not a modification to the manuscript but rather to your rebuttal letter. In particular, please provide a modified, new letter as soon as possible. This should include all of the information you had plus an expanded discussion of reviewer #3’s comments, including the ones that you did not address. Please state how you diverge from or disagree with those comments that you did not address in your revised manuscript. Thank you.

---

## Round 0.3 · accepted · Accept

I believe now the authors have addressed all of the reviewers comments and the manuscript is now ready for publication.

The Section Editor noticed that your response letter does not indicate where responses to Reviewer 3 begin (it does this for Reviewers 1& 2). If you choose to make the review history public, please provide an updated response letter with a heading to make this clear.